# SARS-CoV-2 in Animal Companions: A Serosurvey in Three Regions of Southern Italy

**DOI:** 10.3390/life13122354

**Published:** 2023-12-16

**Authors:** Angelica Bianco, Alessio Bortolami, Angela Miccolupo, Roldano Sottili, Paola Ghergo, Stefano Castellana, Laura Del Sambro, Loredana Capozzi, Matteo Pagliari, Francesco Bonfante, Donato Ridolfi, Carmela Bulzacchelli, Anna Giannico, Antonio Parisi

**Affiliations:** 1Istituto Zooprofilattico Sperimentale della Puglia e Basilicata, Via Manfredonia n. 20, 71121 Foggia, Italy; angelica.bianco@izspb.it (A.B.); stefano.castellana@izspb.it (S.C.); laura.delsambro@izspb.it (L.D.S.); loredana.capozzi@izspb.it (L.C.); donato.ridolfi@izspb.it (D.R.); carmela.bulzacchelli@izspb.it (C.B.); anna.giannico@izspb.it (A.G.); antonio.parisi@izspb.it (A.P.); 2Department of Comparative Biomedical Sciences, Istituto Zooprofilattico Sperimentale delle Venezie, Viale dell’Università 10, 35020 Legnaro, Italy; abortolami@izsvenezie.it (A.B.); mpagliari@izsvenezie.it (M.P.); fbonfante@izsvenezie.it (F.B.); 3ACV Triggiano Laboratorio di Analisi Cliniche Veterinarie, Via Suor Marcella Arosio 8, 70019 Triggiano, Italy; roldano.sottili@izspb.it (R.S.); p.ghergo@acvtriggiano.it (P.G.)

**Keywords:** SARS-CoV-2, COVID-19, ELISA, animals

## Abstract

Several animal species have been found to be susceptible to SARS-CoV-2 infection. The occurrence of infection in dogs and cats living in close contact with owners deserves particular attention from public health authorities in a One Health approach. In this study, we conducted serological screening to identify SARS-CoV-2 exposure in the sera from dogs and cats in three regions of southern Italy sampled during the years 2021 and 2022. We collected 100 serum samples in 2021 (89 from dogs and 11 from cats) and 640 in 2022 (577 from dogs and 63 from cats). Overall, the ELISA positivity rate was found to be 2.7% (20/740), with higher seroprevalence in dogs. Serum neutralization tests confirmed positivity only in two samples collected from dogs, and the assays, performed with serologically distinct SARS-CoV-2 variants, showed variant-specific positivity. This paper shows that monitoring SARS-CoV-2 exposure in animals might be affected by the viral antigenic evolution, which requires continuous updates to the serological tests used. Serological surveys are useful in understanding the true extent of exposure occurring in specific animal populations, not suffering the same limitations as molecular tests, and could help in identifying the infecting virus if tests able to characterize the immune response are used. The use of variant-specific validated serological methods should always be considered in serosurvey studies in order to determine the real impact of emerging variants on animal populations and its implications for veterinary and human health, as well as to identify potential reservoirs of the virus and its evolutionary changes.

## 1. Introduction

Since the spread of the Severe Acute Respiratory Syndrome Virus 2 (SARS-CoV-2), the betacoronavirus (β-CoV) responsible for the global COVID-19 pandemic, there has been an increased interest in coronaviruses (CoVs) in animal populations [1]. These are positive-sense single-strand RNA viruses, whose genome is encoded with four structural proteins, namely spike (S), membrane (M), envelope (E) and nucleocapsid (N) and sixteen non-structural proteins [2]. Emerging CoVs responsible for pandemics or epidemics in humans in the last few decades are phylogenetically related to viruses previously identified in animals, specifically in livestock or wildlife [3,4,5,6]. However, compared to other human CoVs (hCoVs), no clear links between viruses circulating in animal populations and SARS-CoV-2 have been identified, although the highly shared identity with a previously detected bat coronavirus could suggest a bat origin [7,8]. This highlights the need for a deeper understanding of the evolution and interspecific transmission of coronaviruses from animal reservoirs. Recent studies [9,10,11] have shown that different animals, including companion, farm, and wild animals, can be susceptible to SARS-CoV-2. In Italy, as in many other countries, the close relationship between humans and companion or farm animals is common, thus offering ample opportunities for exposure of these animals to SARS-CoV-2 through contact with infected humans, which can result in reverse zoonosis [12,13]. Although there is no evidence that domestic animals are playing a role in the spread of SARS-CoV-2, there is evidence that supports the transmission of the virus from animals to humans [14,15] as a result of sharing the same space, contact, or activities carried out together. Currently, several methods [16] are available for identifying the presence of the virus in different biological matrices that are also of animal origin. However, it is advantageous to carry out serological tests to better estimate the frequency of infection in a specific animal population due to the longer period (i.e., months) of persistence of specific antibodies compared to the presence of the virus, which is usually detectable for a very short period (i.e., days). Among these, ELISAs are essential and critical tools for rapid and economic screening, useful for defining previous exposure and determine seroprevalence in a population [17]. The plasma/serum of collected samples is screened to detect total IGs against a protein of a pathogen for which there is suspected exposure. With the current COVID-19 pandemic, several ELISAs have been developed and made for the detection and quantification of SARS-CoV-2 antibodies [18]. To investigate the SARS-CoV-2 seroprevalence in pets, we screened 740 samples by ELISA based on the recombinant SARS-CoV-2 N protein using a commercially available multi-species ELISA kit. To confirm the results obtained, we then performed seroneutralization assays using antigenically different SARS-CoV-2 variants [19] and detected antibody responses specific to diverse SARS-CoV-2 strains. 

## 2. Materials and Methods

### 2.1. Sample Collection

Animal serum samples were collected from cats and dogs by veterinary practitioners in three regions of southern Italy: Puglia, Basilicata, and Calabria. The blood collection was performed by private veterinary surgeons during routine activities and sent to the ACV Laboratory Srl, which is characterized by a highly specialized team. The laboratory is located in Bari, Apulia, and this explains the abundance of samples from the Apulia region collected in this study. In total, we obtained 740 samples, including 666 from dogs and 74 from cats (Table 1). 

### 2.2. Enzyme-Linked Immunosorbent Assay (ELISA) 

The immunoglobulins and fraction thereof against the N protein of SARS-CoV-2 were measured in all serum samples collected using an ID Screen^®^ SARS-CoV-2 Double Antigen Multi-species ELISA kit (ID.vet, Grabels, France). The choice of the assay was based on the commercial availability of a kit for veterinary use, the abundant literature available that supports the choice of N as a target for screening purposes in animal populations [20], and the superior antigenic stability of the nucleocapsid protein over the spike or the receptor-binding domain (RBD), which are subjected to higher variability, which has possible repercussions for the performance of serological screening tests. Briefly, 20 µL of each serum and control sample was diluted in 30 µL of dilution buffer and incubated for 1 h at room temperature. Plates were washed three times with 300 µL of washing buffer 10×. A total of 50 µL of conjugate dilution buffer 10X was added and incubated for 1 h at room temperature; this wash was repeated three times. The reaction was developed by adding 50 µL of 3,3′,5,5′-tetramethylbenzidine (TMB), and, after an incubation for 20 min at room temperature, the reaction was stopped by adding 50 µL of stop solution. Plates were analyzed with a Thermo Scientific™ Multiskan™ FC Microplate Photometer (ThermoFisher Scientific, Waltham, CA, USA) at an optical density (O.D.) of 450 nm. The absorbance of each sample was obtained by subtracting the O.D. value of the negative control. The cut-off of ≥1 O.D was established for the positive control, and the cut-off of ≤1 O.D was established for the negative control. Results are expressed as percentage of reactivity versus the positive control, calculated as follows: % positivity = (O.D. sample/O.D. mean of positive control) × 100. The samples were considered positive if the percentage was ≥10% and negative if the percentage was ≤10%. The assay was conducted in duplicate for 88% (657/740) of the samples. For the remaining 12% of samples, it was not possible to conduct the assay in duplicate due to insufficient material. Considering that we found no false positives or false negatives for the tests conducted in duplicate, we decided to also include the samples whose material was only assayed once. The O.D. values reported represent the average of the values obtained from the assays.

### 2.3. Focus Reduction Neutralization Test (FRNT)

ELISA-positive serum samples were sent for confirmation to the Department of Comparative Biomedical Sciences of Istituto Zooprofilattico Sperimentale delle Venezie (FAO Reference Centre for Zoonotic Coronaviruses). To confirm the results obtained from ELISA, a focus reduction neutralization test (FRNT) was performed as previously described [21] for the detection of neutralizing antibodies against antigenically distinct SARS-CoV-2 variants with minor modifications. In brief, the live SARS-CoV-2 viruses used in the assays belonged to lineage B.1.617.2 (Delta variant) and BA.2 (Omicron variant), selected on the basis of the epidemiological situation at the time of sampling and due to their antigenic relationship with other variants circulating during the sampling period. SARS-CoV-2 viruses belonging to lineage B.1.617.2 (Delta variant) or BA.2 (Omicron variant) are antigenically distinct variants characterized by approximately a four-fold change in neutralizing activity [22]. Staining of foci was obtained using a specific SARS-CoV/SARS-CoV-2 nucleocapsid monoclonal antibody (Sinobiological Europe GmbH, Eschborn, Germany) and peroxidase-conjugated goat antimouse IgG (Jackson ImmunoResearch, Ely, UK). Foci were visualized on BioSpot™ (CTL Europe GmbH, Rutesheim, Germany). We defined the serum neutralization titer as the reciprocal of the highest dilution that resulted in a reduction in the control focus count higher than 50% (FRNT50). FRNT assays were performed under Biosafety Level 3 conditions.

### 2.4. ELISA-Based Surrogate SARS-CoV-2 Virus Neutralization Test

To obtain further confirmation of the specificity of the immune response detected, FRNT-positive serum samples were tested using a surrogate virus neutralization test (sVNT) (GenScript cPass™ SARS-CoV-2 Neutralization Antibody Detection Kit, Genscript, Rijswijk, The Netherlands), performed according to the manufacturer’s instructions. Briefly, serum samples as well as assay controls were diluted 1:10 in sample dilution buffer and mixed with an equal volume of HRP-conjugated RBDs (derived from wild-type and Omicron variants). Controls and samples were tested in duplicate for both variant-specific versions of the kit. After a 30 min incubation at 37 °C, 100 µL of this mixture was transferred to a 96-well plate coated with recombinant ACE2. After incubation at 37 °C for 15 min, the supernatant was removed and the plate was washed 4X using the provided wash buffer. A total of 100 µL of tetramethylbenzidine substrate was added and incubated for 15 min at room temperature before the reaction was stopped by adding 50 µL of stop solution. Plates were read at 450 nm immediately afterward. Percentage reduction (%reduction) for each sample was calculated by using the following formula: %reduction = (1 − OD_450_(sample)AverageOD_450_(ne.ctrl)) × 100. Samples with values above 30% were considered positive.

### 2.5. Statistical Analysis

Statistical analysis on count data was performed with R version 4.2.2., using the “prop.test” function that calculates Pearson’s chi-squared test statistics with Yates’ continuity correction. This simple function can be implemented for testing the null hypothesis that proportions between two or several groups are the same or that they equal an expected theoretical value.

## 3. Results

Samples were collected from a total of 666 dogs and 74 cats in the period between January 2021 and November 2022 from eleven provinces in southern Italy (Puglia, Basilicata and Calabria regions) (Figure 1; Table 1). Most of the samples were obtained from dogs (90%). With respect to the location, most of the sampled animals were from Bari (Figure 1), which is the closest area to our competence. Among the 740 animals screened using ELISA (the OD values are shown in Figure 2), 20 animals (2%), including 19 (95%) dogs and 1 (5%) cat, had positive test results. No significant differences emerged between canine and feline global ELISA positivity rates (Pearson’s chi-squared test with Yates’ continuity correction, *p*-value = 0.705). The results are summarized in Table 1. A comparison of SARS-CoV-2 ELISA positivity rates by location showed that the virus was detected with a higher prevalence in Puglia (85%), followed by Basilicata (10%) and Calabria (5%). The interprovince SARS-CoV-2 infection rate was comparable (Pearson’s chi-squared test for count data). Seventeen ELISA-positive samples (three samples, including the sample from the cat, were not tested using FRNT due to the insufficient amount of serum) were evaluated for the presence of neutralizing antibodies by FRNT, and two samples were confirmed positive with a low neutralization titer of 1:40 and 1:20, respectively (Table 1). Specifically, one sample was positive for Delta B.1.617.2 and one was positive for Omicron BA.2 variants, as shown by the FRNT results. Both samples also showed positive results with the cPass™ kit, either using the wild-type (FRNT Delta-positive sample) or the Omicron BA.2 (FRNT BA.2-positive sample) HRP-conjugated RBDs.

## 4. Discussion

In the months following the outbreak of the COVID-19 pandemic, several studies were conducted with the aim of understanding the transmission of SARS-CoV-2 to domestic and wild animals [8,10,23,24]. The present study aimed to investigate the serological prevalence of SARS-CoV-2 among companion animals (cats and dogs), mainly from the Puglia region and, due to the collaboration of private veterinary centers, also from two other southern Italian regions (Basilicata and Calabria) through passive surveillance. From January 2021 to November 2022, 740 pets were sampled, and the presence of SARS-CoV-2-specific antibodies was detected using ELISA. The serological survey of antibodies against SARS-CoV-2 allowed us to monitor the spread of the virus among domestic animals, showing a positivity of 0.27% for the samples. Although there was a difference in the sampling during the two years, if we consider the incidence of positive samples as a function of time, there was a higher rate of positivity in 2021 than in 2022. If we take into account the spread trend of the SARS-CoV-2 variants, in 2021, the dominant variant in Italy, as well as in the regions mentioned in the study, was the Delta variant [25], whereas in 2022, the Omicron variant dominated. While it is true that there is no scientific evidence, our results seem to support the thesis that dogs and cats are less susceptible to the Omicron variant [26]. Further studies and investigations are needed to corroborate these findings [27]. Thus, we decided to perform confirmatory tests using two SARS-CoV-2 variants frequently found in humans in Italy during the sampling period that are characterized by a considerable antigenic distance (above four antigenic units, according to van der Straten et al., 2022 and Mykytyn et al., 2022) [28,29]. Only two sera of dogs were confirmed initially via FRNT and then via sVNT: one sampled during 2021 was positive for the Delta variant, and one sampled during 2022 was positive for the BA.2 variant. This result is consistent with the epidemiological trend in the SARS-CoV-2 variants observed in humans during the same period in Italy (Figure 3) [11,30,31], but we cannot exclude positivity owing to infection with antigenically similar variants, such as Alpha (for RFNT Delta-positive sample) or BA.1 and BA.5 (for FRNT Omicron BA.2-positive sample). The samples that tested negative do not question the performance of the ELISA test, whose specificity and sensitivity have been assessed by the manufacturer, but rather highlight the need to monitor the performance of available serological tests for SARS-CoV-2 according to the emergence of new variants and the need to update them given the epidemiological context. A limitation of this study was the different proportions of the dogs and cats sampled. Therefore, finding more SARS-CoV-2 positivity in dogs than in cats may not be related to a greater susceptibility of one species over another, as demonstrated by other authors [9,10,32]. Furthermore, due to privacy restrictions, we do not know the owners of the animals, so we cannot speculate on the route of infection of the dogs that tested positive. However, this study highlights the importance of updating screening and confirmatory tests used in serosurveys and tailoring the assay panel to the collection date of samples to ensure high sensitivity and the reliability of the results.

## 5. Conclusions

In conclusion, we found two SARS-CoV-2-positive dogs using ELISA and FRNT tests, suggesting that the virus had circulated among companion animals, even at a very low rate, in two of the three regions in southern Italy included in the study. Nevertheless, the need to continuously update serological tests is evident from the results obtained in this study, where, due to the specificity of the antibody response to the neutralizing epitopes of the virus, different assays were needed to confirm ELISA-positive serum samples. Updating screening and confirmatory tests is therefore of the utmost importance for accurately estimating the true extent of infections in animal populations. In this study, it was not possible to determine whether the pets were positive for SARS-CoV-2 due to exposure to the infection within the household; in fact, we had no clinical information regarding the COVID-19 disease status of the owners or on the clinical condition of the pets included in the study. Obtaining this information could help enrich scientific observations about the transmission of the virus from humans to animals and vice versa.

## Figures and Tables

**Figure 1 life-13-02354-f001:**
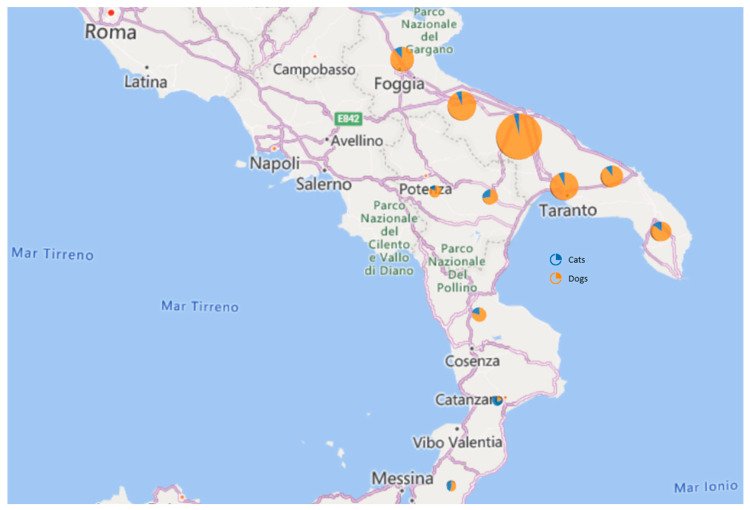
Geographical distribution of sample animals. Each circle represents the geographical area where animals were counted and lived. The size of the area of the circles is proportional to the number of samples collected per area. The cats are indicated with a blue color, and the dogs are indicated with an orange color.

**Figure 2 life-13-02354-f002:**
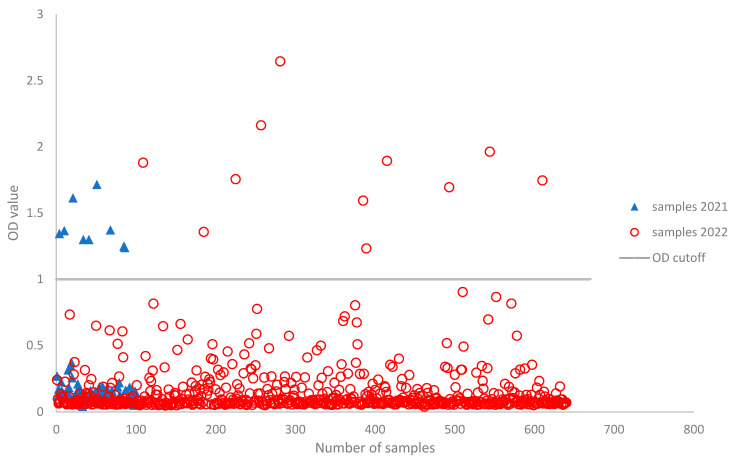
Scatter plot for OD values of ELISAs. The vertical line represents the positive or negative discriminatory cut-off (OD ≥ 1).

**Figure 3 life-13-02354-f003:**
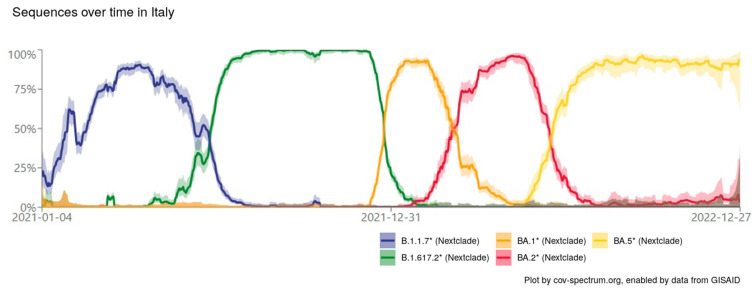
Epidemiological situation in Italy between January 2021 and December 2022, with the percentages of sequences belonging to different variants in that period. In particular, SARS-CoV-2 variants considered in this study are colored in green (B.1.617.2 Delta) and in red (Omicron BA.2). Lines represent the proportion of each variant relative to all samples collected, with shaded areas delimiting the 95% confidence interval. Genomic data are available on the GISAID platform.

**Table 1 life-13-02354-t001:** Samples information and serological results of the enrolled animals. For the positive samples, the title of FRNT and type of CoVs are reported. BA = Bari; BT = Barletta; FG = Foggia; BR = Brindisi; TA = Taranto; LE = Lecce; PZ = Potenza; MT = Matera; CS = Cosenza; CZ = Catanzaro; RC = Reggio Calabria; D/C = dog/cat.

			2021						2022					
		Samples (*n* = 740)	Samples		ELISA		PRNT Assay		Samples		ELISA		PRNT Assay	
Region	Province		Dogs (*n* = 89)	Cats (*n* = 11)	Positive (D/C)	Negative	Positive	Negative	Dogs (*n* = 577)	Cats (*n* = 63)	Positive (D/C)	Negative	Positive	Negative
Puglia	BA	265	36	2	3(3/0)	35	1 (1:40); Delta	2	213	14	3 (3/0)	259	-	5
	BT	104	16	4	2(2/0)	18	-	2	74	10	1 (1/0)	101	-	3
	FG	72	9	1	1(1/0)	9	-	1	55	7	2 (1/1)	69	-	3
	BR	59	8	0	0	8	-		46	5	0	59	-	-
	TA	98	8	1	1(1/0)	8	-	1	80	9	2 (2/0)	95	-	3
	LE	49	4	1	1(1/0)	4	-	1	40	4	1 (1/0)	47	-	2
Basilicata	PZ	18	2	1	0	3	-	3	11	4	1 (1/0)	17	-	1
	MT	29	3	1	0	4	-	4	19	6	0	29	-	-
Calabria	CS	25	1	0	1(1/0)	0	-	-	23	1	1 (1/0)	23	1 (1:20); Omicron BA.2	1
	CZ	10	1	0	0	1	-	1	8	1	0	10	-	-
	RC	11	1	0	0	1	-	1	8	2	0	11	-	-

## Data Availability

Data are contained within the article.

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
