# Peer review of "SARS-CoV-2 in Animal Companions: A Serosurvey in Three Regions of Southern Italy"

_life, 2023, doi:10.3390/life13122354_

Round 1

Reviewer 1 Report

Comments and Suggestions for Authors

The article “SARS-CoV-2 in animal companions: a serosurvey in Southern Italy” by Bianco et al, reports a seroprevalence of SARS-CoV-2 in some region of the south of Italy in cats and dogs.

From a technical point of view the Authors use the best methods to support their findings.

Major revision

The Authors intend to make a seroprevalence of companion animals (only cats and dogs) for the presence of antibody against SARS-CoV-2. The major weakness of the manuscript is the sampling design. Authors intended to test seroprevalence of companion animals from Southern Italy however South Italy is one of the five official statistical regions of Italy used by the Italian Institute of Statistics. South Italy should encompasses at least six of the country's 20 regions.

Moreover the samples appears to have been collected mainly from a single Italian Region “Puglia” 87% and mostly from one province BA (35%). So there are very high difference in sampling between regions and also between provinces.

Minor revision

Line 45 I Guess that Reference 11 is not appropriate

Line 50 This reference is not dealing with SARS-CoV-2

Line 123-125 Are the provinces of samples collection representing the entire region area?

Line 156 Please explain the different intensity of the colors in the Figure.

Author Response

We thank the reviewer for his/her suggestions to improve our article. We have therefore modified the text to better meet the reviewer's requests. Below are our comments on each of his remarks. 

Reviewer 1

The article “SARS-CoV-2 in animal companions: a serosurvey in Southern Italy” by Bianco et al, reports a seroprevalence of SARS-CoV-2 in some region of the south of Italy in cats and dogs.

From a technical point of view the Authors use the best methods to support their findings.

Major revision

The Authors intend to make a seroprevalence of companion animals (only cats and dogs) for the presence of antibody against SARS-CoV-2. The major weakness of the manuscript is the sampling design. Authors intended to test seroprevalence of companion animals from Southern Italy however South Italy is one of the five official statistical regions of Italy used by the Italian Institute of Statistics. South Italy should encompasses at least six of the country's 20 regions.

Moreover the samples appears to have been collected mainly from a single Italian Region “Puglia” 87% and mostly from one province BA (35%). So there are very high difference in sampling between regions and also between provinces.

We thank the reviewer for the comment. We share the reviewer's thoughts on sampling. In a passage in the results section (in the new version Line 151) we briefly explained that the abundance of samples in the Province of Bari is justified by the nearness of that Province to our area, which is closely related to the Clinical Veterinary Laboratory that collected the individual samples. In order to better specify this aspect, we have modified the text. We have also modified the text of the manuscript with regard to the southern Italian regions involved.

Minor revision

Line 45 I Guess that Reference 11 is not appropriate

We thank the reviewer, there must have been an 'error' in entering the reference. We modified the reference citation.

Line 50 This reference is not dealing with SARS-CoV-2

We thank the reviewer and we modified the reference citation.

Line 123-125 Are the provinces of samples collection representing the entire region area?

All the provinces in the Apulia and Basilicata regions and the larger ones in the Calabria region are involved in the study. Certainly, for the Apulia Region the sampling is larger and therefore more representative of the spread of the virus in the domestic animals analysed.

Line 156 Please explain the different intensity of the colors in the Figure.

Based on the reviewer's comment, we modified the caption.

Reviewer 2 Report

Comments and Suggestions for Authors

The submitted manuscript entitled sars-cov-2 in animal companions: a serosurvey in Southern Italy by Bianco et al, has been reviewed. Authors investigated the serosurvey in dog and cat sera that were obtained during 2021-2022 with passive sample collection. Within the 740 sera, 20 sera were seropositive by ELISA. The positive samples (only 17 sera) were then submitted for serum neutralization, which 2 samples were positive. Authors indicated that viral antigenic evolution may result in alteration of the positive rate. Authors also raised the serological test should be updated due to antigenic changes. 
Overall, this manuscript found evidence support the SARS-CoV-2’s exposure/ infection of dogs and cats, which have been reported elsewhere, and may support other previous findings. There are several points that I expected to be responded by the authors, together with the questions in several aspects, and I will describe collectively. 

1. L23. Authors indicated that “ Serological surveys are useful to 22 understand the true extent of infections occurred in specific animal populations,…” seropositive is notonly indicate the infection, but also may due to exposure. Authors should determine that this is exposure or infection, unless do other test to confirm the infection.

2. L49. Infection of SARs- CoV-2 in dogs and cats is not only reverse zoonosis. There are reports indicate the possible zoonosis of SARs-CoV-2 infection from dogs and cats. Authors should obtain the information to be included within this aspect. 
3. L66: Sample collection. I do recognize that authors passively obtained the samples from the Vet clinics; however, do authors have specific or inclusive criteria for samples collections as I can find that the most samples were obtained in 2022, much higher than 2021. How the samples were kept? Long storage or hemolysis (even I don’t know) may alter the ELISa. 
4. L72: Authors measured IgG against N protein of SARs. I do understand regarding this, but it would be better to indicate why is N, instead of RdRp or S protein, even within here or indicate the importance of N protein in the Introduction, to convince reader why measurements of anti-N IgG. 
L71-87: with the respect of performance of the ELIZA, authors should duplicate or triplicate the results obtained from the ELISA, OR Authors have specific conditions why performed only one time (should indicate). Together with negative results, it may be false or true; if authors do duplicate can convince the result. 
L94-95: Authors performed the FRNT using delta and BA.2 variants following the epidemiological situation in such area, and the results were confirmed in only 2 cases, so authors merit the findings that the serological test should be updated due to antigenic changes of the virus. Authors should have detailed in this context for non-field reader ( as I can find in the result section. 
L121: Although the used statistics have beeb indicTed in the results section ( ie Person?and others), in term of method authors should indicate clearly what is each statistics used for? This 

Figure 1: Authors should label the name of the city or province to better understand in non-Italian reader by do not read this paper together with flipping the map. 
Figure 2: I have noticed that the proportion of the positive samples/ collected samples is higher in 2021 during the Delta’s prevalence. As many studies have found that the BA has less infectivity to dogs and cats, compared to other VOC, where authors lack in this discussion. 

Table1. Representative data are difficult to read due to falling paragraph of the info, please correct. 

Discussion: Authors discussed that “ the need of a continuous update of serological tests is evident from the results obtained in this study, where, due to the specificity of the antibody response to the neutralizing epitopes of the virus, different assays were needed to confirm ELISA positive serum samples. ” i am agreeing with this point whereas there are other points need to be discussed, not only the test. 

Minor: 

L180: Rice? 
L181: SRS? 

Comments on the Quality of English Language

Use of English language is acceptable. There are some typos presented. 

Author Response

We thank the reviewer for his/her suggestions to improve our article. We have therefore modified the text to better meet the reviewer's requests. Below are our comments on each of his remarks. 

Reviewer 2

The submitted manuscript entitled sars-cov-2 in animal companions: a serosurvey in Southern Italy by Bianco et al, has been reviewed. Authors investigated the serosurvey in dog and cat sera that were obtained during 2021-2022 with passive sample collection. Within the 740 sera, 20 sera were seropositive by ELISA. The positive samples (only 17 sera) were then submitted for serum neutralization, which 2 samples were positive. Authors indicated that viral antigenic evolution may result in alteration of the positive rate. Authors also raised the serological test should be updated due to antigenic changes. 
Overall, this manuscript found evidence support the SARS-CoV-2’s exposure/ infection of dogs and cats, which have been reported elsewhere, and may support other previous findings. There are several points that I expected to be responded by the authors, together with the questions in several aspects, and I will describe collectively. 

  1. Authors indicated that “ Serological surveys are useful to 22 understand the true extent of infections occurred in specific animal populations,…” seropositive is notonly indicate the infection, but also may due to exposure. Authors should determine that this is exposure or infection, unless do other test to confirm the infection.

We thank the reviewer for the comment. Based on the data obtained and reported in the manuscript, we agree to change the sentence to speak of exposure rather than infection.

  1. Infection of SARs- CoV-2 in dogs and cats is not only reverse zoonosis. There are reports indicate the possible zoonosis of SARs-CoV-2 infection from dogs and cats. Authors should obtain the information to be included within this aspect. 

We thank the reviewer; in the revision we included this aspect.

  1. L66: Sample collection. I do recognize that authors passively obtained the samples from the Vet clinics; however, do authors have specific or inclusive criteria for samples collections as I can find that the most samples were obtained in 2022, much higher than 2021. How the samples were kept? Long storage or hemolysis (even I don’t know) may alter the ELISa. 

The samples as described were collected during the two-year: 2021-2022. We specify that they were collected throughout the described time frame. However, the study foresaw an increase in the number of samples in the year 2022 by expressly asking the veterinary centres to send more samples. it is evident that the laboratories responded positively to this request. this justifies the difference in the number of samples received in the two years. We received the frozen samples and followed this preservation protocol:

  • if the samples were analysed immediately, thaw the serum and process it;
  • if samples are not tested on arrival, store them at -20°C (up to one month) or -80°C (up to two months);
  • avoid freeze-thaw cycles. Bring samples to room temperature before carrying out the assay.
  1. L72: Authors measured IgG against N protein of SARs. I do understand regarding this, but it would be better to indicate why is N, instead of RdRp or S protein, even within here or indicate the importance of N protein in the Introduction, to convince reader why measurements of anti-N IgG. 

We modified the text in material e method section opportunely.

L71-87: with the respect of performance of the ELIZA, authors should duplicate or triplicate the results obtained from the ELISA, OR Authors have specific conditions why performed only one time (should indicate). Together with negative results, it may be false or true; if authors do duplicate can convince the result. 

The assay was carried out in duplicate for 100 % of samples collected on 2021 and for ~ 87% (557/640) of samples collected on 2022. The results reported in the article are the average values obtained from the assays. Unfortunately, this was not possible for some because the material was not sufficient. Based on the duplicate assays and on previous tests conducted, we did not find any false-positive or false-negative samples. Additionally, for each assays the controls were always tested in duplicate.  We therefore decided to present all the samples analysed even though for some it was not possible to perform the assay twice. We have therefore integrated this information into the text.

L94-95: Authors performed the FRNT using delta and BA.2 variants following the epidemiological situation in such area, and the results were confirmed in only 2 cases, so authors merit the findings that the serological test should be updated due to antigenic changes of the virus. Authors should have detailed in this context for non-field reader ( as I can find in the result section. 

We modified the text opportunely.

L121: Although the used statistics have beeb indicTed in the results section ( ie Person?and others), in term of method authors should indicate clearly what is each statistics used for? This 

Thanks for the remark. Details of the used statistical function has been added in lines 142-145

Figure 1: Authors should label the name of the city or province to better understand in non-Italian reader by do not read this paper together with flipping the map.

We thank for this suggestion; we modified the text.  

Figure 2: I have noticed that the proportion of the positive samples/ collected samples is higher in 2021 during the Delta’s prevalence. As many studies have found that the BA has less infectivity to dogs and cats, compared to other VOC, where authors lack in this discussion. 

We thank the reviewer for the suggestion. We have edited the text in the discussion section accordingly.

Table1. Representative data are difficult to read due to falling paragraph of the info, please correct. 

We agree with the reviewer's opinion. We have adhered to the guide for authors which requires tables to be inserted as text and not as images. For these reasons, the table is difficult to understand. We believe that if the manuscript is to be published, it will be up to the reviewer to adapt the table for correct and easy viewing. Otherwise, we will ask for it to be adapted. To help the reviewer, we provide an image of the table itself below.

Reviewer 3 Report

Comments and Suggestions for Authors

It is an interesting work.

OD levels appear to be close to the Cut off limit. Could there be cross or false positivity?

Are the animals from which blood is collected healthy? Were they brought to the vet for a check-up or because they were sick?

Have cats been previously tested for other Corona viruses (FIP??)? It may be meaningful to consider the possibility of cross-positivity.

Has a clinical condition with respiratory tract infection or GE findings been identified in these animals during or after the pandemic period?

Is the covid-19 history of the household known?

The discussion section could be written in more detail.

It may be meaningful if the conditions under which it passes to pets in humans, whether it originates from humans or another pet, and the state of transmission from pets to humans can be examined.

Care should be taken in such studies and interpretations, and pads should not be mistakenly perceived as reservoirs or sources. This is a very sensitive point and should be clearly emphasized in the conclusion of the article

Author Response

We thank the reviewer for his/her suggestions to improve our article. We have therefore modified the text to better meet the reviewer's requests. Below are our comments on each of his remarks. 

Reviewer 3

It is an interesting work.

OD levels appear to be close to the Cut off limit. Could there be cross or false positivity?

We thank the reviewer for the comment. It was not specified in the text before the revision that the assays were largely done in duplicate. Due to insufficient material, it was not possible to perform the assay in duplicate for a few samples. However, in light of the fact that no false positives or negatives were found in the samples tested positive, we thought it would be useful to include those for which the material was insufficient. in response to your comment, we believe that the value is merely close to the cut-off, but we do not consider there to be any problems related to cross o false positivity.

Are the animals from which blood is collected healthy? Were they brought to the vet for a check-up or because they were sick?

We find the remarks made interesting. Unfortunately, we are unable to answer them because we have no clinical information on the animals whose serum was used for research purposes, this for ethical reasons.

Have cats been previously tested for other Corona viruses (FIP??)? It may be meaningful to consider the possibility of cross-positivity.

This is also a good suggestion. We have the resources to be able to investigate the presence of Coranaviruses in suitable samples (PanCoranavirus research, which includes not only FIPV but also Canine Coronvirus). However, the biological material received is not suitable for such an investigation (a respiratory swab would have been very useful).

Has a clinical condition with respiratory tract infection or GE findings been identified in these animals during or after the pandemic period?

As described above, we are unable to answer this question.

Is the covid-19 history of the household known?

As described in the article, we do not know the clinical condition of the owners, just as we do not know the clinical condition of the pets.

However, we also reiterate that it represents a limitation and that it could be useful to also include this information for a more complete serological investigation.

The discussion section could be written in more detail.

Based on the reviewers suggestions, we modified the text.

It may be meaningful if the conditions under which it passes to pets in humans, whether it originates from humans or another pet, and the state of transmission from pets to humans can be examined.

We find this idea very interesting and we share it. However, based on the information we have, it is not possible to delve deeper into this topic, which is also little studied.

Care should be taken in such studies and interpretations, and pads should not be mistakenly perceived as reservoirs or sources. This is a very sensitive point and should be clearly emphasized in the conclusion of the article

In agreement with the reviewer, we modified the text

Round 2

Reviewer 1 Report

Comments and Suggestions for Authors

Dear Editor,

I think that the manuscript is not so interesting for the journal, but the Authors have answered to all my remarks.

Regards

Gabriele

Author Response

Dear reviewer, 
we thank you for your valuable advice and as you have seen, we have taken up each suggestion with interest. However, we regret that you do not find our study interesting. 

Reviewer 2 Report

Comments and Suggestions for Authors

Thanks authors to provide the revision. Changes were made following my suggestions; however, there are minor corrections. Regarding the details of SARS-Cov-2 infection in humans that probably infected from pet cats and dogs. The references you have cites (ref no. 14-15) are not in the line of original information. 

Comments on the Quality of English Language

Authors should rely on the original study of this information. So, I suggested to revise this point. Furthermore, authors have indicated the "Three regions of Italy for sample collection"; however, the word "regions" in this context should be not started by the capital letter. 

Author Response

Dear reviewer,
we thank you for your valuable advice and have further modified the text. 
Cordially
The authors